# Columnar Aggregates of Azobenzene Stars: Exploring Intermolecular Interactions, Structure, and Stability in Atomistic Simulations

**DOI:** 10.3390/molecules26247598

**Published:** 2021-12-15

**Authors:** Markus Koch, Marina Saphiannikova, Olga Guskova

**Affiliations:** 1Institute Theory of Polymers, Leibniz Institute of Polymer Research Dresden, Hohe Str. 6, 01069 Dresden, Germany; grenzer@ipfdd.de; 2Dresden Center for Computational Materials Science (DCMS), Technische Universität Dresden, 01062 Dresden, Germany

**Keywords:** azobenzenes, supramolecular assembly, hydrogen bonding, molecular dynamics, computer simulations

## Abstract

We present a simulation study of supramolecular aggregates formed by three-arm azobenzene (Azo) stars with a benzene-1,3,5-tricarboxamide (BTA) core in water. Previous experimental works by other research groups demonstrate that such Azo stars assemble into needle-like structures with light-responsive properties. Disregarding the response to light, we intend to characterize the equilibrium state of this system on the molecular scale. In particular, we aim to develop a thorough understanding of the binding mechanism between the molecules and analyze the structural properties of columnar stacks of Azo stars. Our study employs fully atomistic molecular dynamics (MD) simulations to model pre-assembled aggregates with various sizes and arrangements in water. In our detailed approach, we decompose the binding energies of the aggregates into the contributions due to the different types of non-covalent interactions and the contributions of the functional groups in the Azo stars. Initially, we investigate the origin and strength of the non-covalent interactions within a stacked dimer. Based on these findings, three arrangements of longer columnar stacks are prepared and equilibrated. We confirm that the binding energies of the stacks are mainly composed of π–π interactions between the conjugated parts of the molecules and hydrogen bonds formed between the stacked BTA cores. Our study quantifies the strength of these interactions and shows that the π–π interactions, especially between the Azo moieties, dominate the binding energies. We clarify that hydrogen bonds, which are predominant in BTA stacks, have only secondary energetic contributions in stacks of Azo stars but remain necessary stabilizers. Both types of interactions, π–π stacking and H-bonds, are required to maintain the columnar arrangement of the aggregates.

## 1. Introduction

The formation of supramolecular aggregates is governed by the non-covalent interactions between their molecular constituents. Changes in the chemical composition of the constituent molecules modify their mutual interactions, leading to different arrangements of the molecular building blocks upon aggregation. The type of arrangement, then, affects the structure, shape, size as well as other chemical and physical properties of the resulting materials [1]. It is, thus, not surprising that supramolecular aggregation phenomena are found in countless biological systems [2,3,4].

Beyond nature, the research into supramolecular assembly has rapidly progressed in recent years, enabling the design of materials with precisely tuned properties and functionalities [5,6]. The applications of supramolecular structures are as multifaceted as the underlying molecules. Such structures are utilized, for instance, in gels [7] and sensing devices [8]. Relatively little-exploited but very promising is the usage of supramolecular systems as adhesives [6,9], in medical and drug delivery applications [6,10], as synthetic biomaterials [11], and in devices to probe or manipulate biological systems [3,12].

A widely employed and studied molecule in supramolecular chemistry is benzene-1,3,5-tricarboxamide (BTA) [13,14,15,16,17,18,19,20]. Due to its oblate shape and specific chemical structure, BTA forms columnar aggregates. Pairs of these $C_{3}$-symmetrical molecules can form up to three hydrogen bonds between their amide groups. The column-shaped molecular stacks of BTAs are typically characterized by a helical arrangement of the molecules and a triple-H-bonding pattern throughout the entire aggregate [14,15,16,18]. BTAs are used to synthesize supramolecular polymers [21,22], act as gelating agents [23,24], and are employed in different medical applications [25,26]. Moreover, these molecules have promising applications as organic ferroelectric materials, e.g., for memory devices [27], and for the design of synthetic biopolymers [11].

The progression of the self-assembly and the structure of the supramolecular aggregates can be directly influenced by the substitution pattern of the BTA scaffolds [28,29,30,31,32]. In particular, the substituents modify the shape and the intermolecular interactions of the molecules. On top of that, adding specific functional groups can even introduce stimuli-responsive behavior to the resulting BTA derivatives and their assemblies. Attaching one or several molecular photoswitches, such as azobenzene (Azo) to a BTA group, yields a molecule that responds to light. This is desirable due to the favorable properties of the light stimulus, such as the possibility of precise control and the absence of waste. Previous studies have investigated BTAs substituted with at least one [32] and up to three Azo units [33,34,35,36,37,38,39,40]. The resulting $C_{3}$-symmetrical molecule, in the latter case, has the appearance of a three-arm star and its three light-reactive Azo groups make it a multiphotochromic compound [33,41,42,43,44].

The formation and the properties of supramolecular aggregates built from such three-arm azobenzene stars have been studied in various experimental [37,38,39,45] and theoretical works [46,47]. It was found that these molecules assemble into different morphologies depending on the solvent [37,38]. For instance, adding water to a solution of three-arm Azo stars in DMSO yields needle-like structures [37], which on the smallest scale consist of columnar stacks of Azo stars. These needle-like structures are light-responsive and undergo an order–disorder transition upon irradiation with UV–Vis light [37]. Hence, aggregates of three-arm Azo stars are promising building blocks for photo-controllable materials like supramolecular polymers and gels.

A thorough comprehension of these aggregates includes a deeper insight into their structural properties, not only in response to light but, perhaps most importantly, first in equilibrium. As already stated in previous experimental studies [37,38], it is clear that π–π interactions contribute to the binding of aggregates composed of such flat aromatic molecules. On top of that, it can be expected that hydrogen bonds are forming between the BTA groups at the centers of the Azo stars. The presence of H-bonds has been confirmed in these earlier experimental works [37,38]. It is unclear, however, which role these interactions play in the binding of the aggregates.

It should be considered that azobenzenes are comparatively large and rigid substituents, which introduce different intermolecular interactions to this BTA derivative. Thus, even aside from the light-induced effect, the Azo arms modify the inner arrangement and the binding mechanism of the resulting supramolecular aggregates. To maximize their π–π interaction, the Azo stars need to stack their conjugated parts, i.e., BTA centers and Azo moieties. By contrast, hydrogen bonding between nearby BTAs requires a specific arrangement of their amide groups, which can be realized if adjacent stacked BTAs are rotated by a relative in-plane angle of $60{}^{\circ }$ [14,15,18]. It remains to be investigated whether the π–π and H-bonding interactions compete with each other due to their dependence on the relative arrangement of the molecules, and whether one of the two interactions dominates. Thus, also the equilibrium structure of the supramolecular aggregates is expected to depend on the interplay of these interactions.

We have previously studied columnar assemblies of three-arm Azo stars using density functional theory and fully atomistic MD simulations [46,47]. In ref. [46], we have simulated the self-assembly of Azo stars from a random distribution in water and analyzed their structure and stability both in equilibrium and in response to light. Ref. [47] has focused on further analyzing the self-assembly mechanism of the Azo stars in water, which was found to proceed cooperatively. In the same work, we have analyzed data of the crystalline phase of the Azo stars, which were previously published by Lee et al. [37]. However, the force field used for our previous MD simulations models hydrogen bonding only implicitly. Thus, a detailed investigation of the different contributions to the binding energies in these aggregates has so far not been carried out.

The present study focuses on the addressed open questions with respect to this system. Our aim is to understand how the functional groups of each Azo star and their different non-covalent interactions contribute to the binding of the aggregates, and how this influences their structures. To this end, we employ fully atomistic MD simulations using a computational model that includes an explicit hydrogen bonding potential. We consider several arrangements of columnar clusters of Azo stars and analyze their stability and binding mechanism. In particular, we carry out a detailed analysis of the intermolecular interactions in the aggregates. The new approach allows us to quantify how the functional groups in the Azo stars and the different types of non-covalent interactions contribute to the binding energies of the aggregates. Thus, the roles and the possible competition between these interactions are examined. Note that this article is based on the dissertation of M.K. [48].

The remainder of this publication is organized as follows. In Section 2, we describe the objects of study, i.e., the three-arm Azo star and its aggregates. On top of that, the utilized simulation models and methods are introduced. In the subsequent Section 3, the results of the study are provided and discussed. We begin by examining the intermolecular energy landscape of a dimer of two stacked azobenzene stars. Thereafter, we characterize the structural properties of long columnar stacks of Azo stars and analyze the intermolecular interactions between the stacked molecules. Section 4 summarizes the findings of this work and provides our conclusions. Appendix A contains results regarding the formation of macrodipoles in stacks of Azo stars and the Appendix A contains additional figures and definitions of numerous observables analyzed in this work.

## 2. Materials and Methods

### 2.1. Object of Study: Supramolecular Aggregates of TrisAzo

At the center of this study is a three-arm azobenzene star, here given the name *TrisAzo* (Figure 1). It consists of three azobenzene groups, which are linked by a BTA group at the center of the compound. Each Azo group carries a dimethylamino (DMA) group, attached at the para position: R = N(CH_3_)_2_. The constituents DMA and BTA are photoinert, whereas the Azo groups can undergo reversible photoisomerization under UV–Vis light. Upon absorption of a photon, azobenzene isomerizes from the stable *trans* to the metastable *cis* state or vice versa. Since *TrisAzo* incorporates three photochromic Azo moieties, it is a multiphotochromic molecule [33,41,42,43,44]. Due to the possible photoisomerization of each Azo arm, four different isomerization states—from all-*trans* to all-*cis*—may be toggled by photon absorption [33]. Since the influence of light on *TrisAzo* is not investigated in the present study, we only consider the molecule in its all-*trans* state. Note that the three-arm azobenzene stars studied in some of the referenced experimental studies carry different peripheral substituents but are otherwise identical to *TrisAzo*. In the case of R = H, we refer to the molecule as *TrisAzo-H* [37], and for R = Br, *TrisAzo-Br* [38].

We consider different types of supramolecular aggregates formed by *TrisAzo* molecules. The preparation of *TrisAzo* dimers is described below (Section 2.4). For the simulation of larger columnar *TrisAzo* aggregates (N≥2 monomers), the molecules are pre-assembled into different stacked arrangements. Details of the considered supramolecular geometries are provided in Section 3.2.

### 2.2. Simulation Model

The supramolecular aggregates of *TrisAzo* are studied via fully atomistic MD simulations. The parameterization of *TrisAzo* molecules is according to the DREIDING force field [49]. Based on the hydrogen bonding potential included in DREIDING, we explicitly model the hydrogen bonds between the BTAs of opposing *TrisAzo* molecules. The default parameters of the H-bonding potential of DREIDING are used [49]. Assignment of the force field parameters and Gasteiger partial charges is carried out in BIOVIA Materials Studio [50]. Dimers and columns of *TrisAzo* are created by depositing molecules at specific locations in the software Moltemplate [51]. These structures are solvated in SPC/E water [52] with the aid of the software PACKMOL [53,54]. All files are transferred to the open-source simulation package LAMMPS, in which the equilibration and production runs are performed [55,56].

Simulations are carried out in the NPT ensemble (T=300 K, P=1atm) to match experimental conditions [37]. To this end, a Nosé–Hoover-style barostat and thermostat as implemented in LAMMPS [55,56] is applied. The Lennard–Jones (LJ) potentials are cut off at 9.8 Å and Lorentz–Berthelot combination rules [57,58] are used for the LJ interactions between dissimilar atoms. For the Coulomb interactions, the PPPM algorithm is used (accuracy of $10^{-4}$) [59,60]. Bond angles and bond lengths of covalent bonds involving hydrogen atoms are constrained by the SHAKE algorithm [61]. The integration step of the simulations is 1 fs.

### 2.3. Calculation of the Intermolecular Energies

In this work, we consider the intermolecular energy within dimers or larger aggregates of molecules. The total intermolecular energy of a group of atoms, $E_{\mathrm{tot}}^{\mathrm{inter}}$, is the sum of all energies arising from interactions of each pair of atoms *I* and *J*, where *I* and *J* are part of two different molecules *i* and *j*. We define $E_{\mathrm{tot}}^{\mathrm{inter}}$ as
(1)$$\begin{array}{c} E_{\mathrm{tot}}^{\mathrm{inter}}=\sum_{I\neq J}E_{\mathrm{Coulomb}}+\sum_{I\neq J}E_{\mathrm{vdW}}+\sum_{I\neq J\neq K}E_{\mathrm{hb}},\;\mathrm{with}\;I\in i,\;J,K\in j,\;i\neq j. \end{array}$$

The energy terms on the right-hand side refer to the different non-covalent energies, i.e., the Coulomb energies $E_{\mathrm{Coulomb}}$, the van der Waals energies $E_{\mathrm{vdW}}$ and the hydrogen bonding energies $E_{\mathrm{hb}}$. Their definitions are provided in Appendix A. In the case of intermolecular hydrogen bonds, three atoms are involved. Equation (Equation 1) implies that the acceptor atom *I* belongs to a different molecule than the donor and hydrogen atom, *J* and *K*. In analogy to Equation (Equation 1), one can evaluate only specific contributions to the total intermolecular energy, e.g., the intermolecular Coulomb energy:(2)$$\begin{array}{c} E_{\mathrm{Coulomb}}^{\mathrm{inter}}=\sum_{I\neq J}E_{\mathrm{Coulomb}},\;\mathrm{with}\;I\in i,\;J\in j,\;i\neq j. \end{array}$$

When one considers only such atoms *I* and *J* that belong to specific parts of the molecules, e.g., certain functional groups, the intermolecular energies can be decomposed further.

### 2.4. Calculation of the Intermolecular Energy Landscape of a TrisAzo Dimer

We compute the intermolecular energy landscape of a stacked *TrisAzo* dimer, i.e., a two-dimensional plot displaying the intermolecular energies between the molecules for different spatial arrangements. To this end, the intermolecular energies of numerous dimer configurations are calculated, each providing a data point to the energy landscape. At each data point, we take the sum of all non-covalent energetic interactions arising between two atoms that are not part of the same *TrisAzo* molecule. A 2D plot results by combing the intermolecular energies of dimer configurations with different center-of-mass (COM) distances, Δr, and in-plane rotation angles, Δφ. The considered COM distances are in the range 1.0 Å≤Δr≤6 Å, in steps of 0.05 Å. Smaller COM separations lead to unphysical overlaps and are omitted. The upper limit corresponds to the first minimum in the radial distribution function in previous self-assembly simulations of *TrisAzo* molecules [46]. The rotation angles are in the range $0{}^{\circ }\leq \Delta \varphi <360{}^{\circ }$, in steps of $1{}^{\circ }$. In total, the energy landscape consists of 101×360=36,360 data points.

Each dimer geometry is created in the software tool Moltemplate [51] by centrally placing two *TrisAzo* molecules into a rectangular simulation box of dimensions $L_{x}\times L_{y}\times L_{z}=80\;Å\times 80\;Å\times 100\;Å$ at the selected values of Δr and Δφ. For the calculation of the energy landscape, solvent molecules are omitted. Here, the LJ interactions are cut off at 15 Å since computing them for this system with higher precision does not significantly diminish the computational performance. Thereafter, energy minimization is carried out using the conjugate gradient algorithm, i.e., the Polak–Ribiere version [62] as implemented in LAMMPS [55,56]. Note that the atoms of the central phenyl rings of both *TrisAzo* molecules are constrained in their spatial position. This ensures that the selected COM distance and rotation angle remain valid. Lastly, all desired intermolecular energy terms are calculated and the full energy landscape is created.

## 3. Results and Discussion

### 3.1. Intermolecular Energy of a TrisAzo Dimer

Previous simulations of *TrisAzo* molecules in water have demonstrated that the molecules self-assemble into column-like aggregates [46]. The highly orientated aggregation corresponds to the needle-like structures observed in the experiments [37,38]. This type of attachment results from the strongly anisometric shape of the molecule (entropic effect), as well as specific non-covalent interactions. On the smallest scale, the columnar cluster is comparable to a chain of *TrisAzo* dimers, in which the planes of the Azo arms are parallel to each other. We first study the intermolecular interactions in such dimers to help predict the structural properties of larger columnar stacks.

Due to the symmetry and central alignment of the dimer geometry, the intermolecular energy of the *TrisAzo* dimers is studied as a function of two variables: the COM distances of the molecules, Δr, and the rotation angle Δφ between them around an axis connecting their COM (Figure 2a). Other degrees of freedom that influence the intermolecular energies are not taken into account in this representation. Our previous self-assembly simulations indicate that such an alignment corresponds to the spontaneously formed dimer structure [46]. The lateral shift between the molecules and their relative inclination angle Δα are set to zero. Let us shortly remark how Δα is computed. Considering the all-*trans* isomers of *TrisAzo* as disk-like, oblate objects, the direction perpendicular to the plane of their arms may be used to define their orientation. To obtain the respective orientation unit vectors of the molecules, we follow an approach suggested in ref. [63]. The angle between the orientation unit vectors of two such molecules is the inclination angle Δα. More details are provided in the Appendix A.

The total intermolecular energy is, therefore, extracted in the form $E_{\mathrm{tot}}^{\mathrm{inter}}\left(\Delta r,\Delta \varphi \right)$. Figure 2b shows $E_{\mathrm{tot}}^{\mathrm{inter}}$ both as a three-dimensional and a two-dimensional plot in cylindrical coordinates to illustrate the energy landscape. Note that only the two-dimensional representation of $E_{\mathrm{tot}}^{\mathrm{inter}}$ will be used further on.

#### 3.1.1. Total Intermolecular Energy of the Dimer

The total intermolecular energy of the dimer is depicted in Figure 3a as a two-dimensional plot in cylindrical coordinates. The $C_{3}$ symmetry of each *TrisAzo* manifests itself in the pattern of repulsive and attractive areas of the energy landscape. Note that the rotation of one *TrisAzo* by $\Delta \varphi =120{}^{\circ }$ or $240{}^{\circ }$ corresponds to the same spatial arrangement as $0{}^{\circ }$. Hence, approximately the same intermolecular energies are found after such rotations. As expected, a strongly repulsive, radially symmetric region with positive energy is found at short COM distances (Δr≲2.5 Å), which results from steric repulsion between closely approaching atoms. This region marks the strongest repulsive interactions. Here, the energy values rise steeply to large positive values beyond the displayed energy scale of $60\;k\mathrm{cal}\;\cdot \;\mathrm{mol}{}^{-1}$ as the COM distance decreases.

Besides this repulsive part, the energy plot reveals overall attractive interactions for Δr≳3.0 Å at all rotation angles. The negative (attractive) energy range has three local minima, at about Δr≈3.7 Å, $\Delta \varphi =0{}^{\circ }$, $120{}^{\circ }$ and $240{}^{\circ }$. The global minimum is $E_{\mathrm{tot}}^{\mathrm{inter}}\left(3.7\;Å,\;0{}^{\circ }\right)=-51.7\;k\mathrm{cal}\;\cdot \;\mathrm{mol}{}^{-1}$, while the others are shallower by ca. $6\;k\mathrm{cal}\;\cdot \;\mathrm{mol}{}^{-1}$. This difference is likely an inaccuracy of the energy minimization procedure since the three minimum values should be equal for symmetry reasons. The local energy minima are at the center of potential wells with an extent of about 1.5 Å by $\pm 20{}^{\circ }$. In the regions around $\Delta \varphi =60{}^{\circ }$, $180{}^{\circ }$ and $300{}^{\circ }$, the attractive energies are weaker and have a slightly irregular pattern. These results imply that approximately linearly stacked *TrisAzo* molecules, i.e., geometries with no or small relative rotations, yield the strongest binding.

#### 3.1.2. Decomposition of the Intermolecular Energy of the Dimer

The question arises of which roles different kinds of interactions play; for instance, which ones are dominant, and how they promote or suppress certain dimer geometries? To this end, we follow the approach suggested in ref. [32] and decompose $E_{\mathrm{tot}}^{\mathrm{inter}}$ into different contributions. In particular, the intermolecular energy can be decomposed into various sets of interactions:the different types of non-covalent interactions, andthe interactions between different parts of the molecule.

For the decomposition of $E_{\mathrm{tot}}^{\mathrm{inter}}$ by the types of non-covalent interactions, the following contributions are considered:$E_{\mathrm{vdW}}^{\mathrm{inter}}$ — van der Waals interactions,$E_{\mathrm{Coulomb}}^{\mathrm{inter}}$ — Coulomb interactions,$E_{\mathrm{hb}}^{\mathrm{inter}}$ — explicit hydrogen bonds.

These contributions sum to the total intermolecular energy. The definition of each energy term is provided in Appendix A.

The decomposed energy landscapes are depicted in Figure 3b–d. Among the three contributions, $E_{\mathrm{vdW}}^{\mathrm{inter}}$ is the most dominant (Figure 3b). It contains both the strongly repulsive interactions at close distances (steric hindrance) and the highly attractive regions at $\Delta \varphi =0{}^{\circ }$, $120{}^{\circ }$ and $240{}^{\circ }$ (London dispersion interactions). The three minima of $E_{\mathrm{vdW}}^{\mathrm{inter}}$ are found at the same coordinates as in $E_{\mathrm{tot}}^{\mathrm{inter}}$ and are even slightly deeper. By comparison, the contributions of $E_{\mathrm{Coulomb}}^{\mathrm{inter}}$ are much weaker (Figure 3c). The Coulomb energies vary from negative to positive values (−6 to $14\;k\mathrm{cal}\;\cdot \;\mathrm{mol}{}^{-1}$) in stripe-like sections, overlaying the attractive regions of $E_{\mathrm{vdW}}^{\mathrm{inter}}$. Lastly, the hydrogen bonding interactions (Figure 3d) yield only attractive contributions ($\min E_{\mathrm{hb}}^{\mathrm{inter}}=-12.7\;k\mathrm{cal}\;\cdot \;\mathrm{mol}{}^{-1}$). Hydrogen bonds are formed at rotation angles around $\Delta \varphi =60{}^{\circ }$, $180{}^{\circ }$ and $300{}^{\circ }$, in accordance with the geometrical requirements of H-bond formation between stacked BTAs [14,15,18]. Even when combined, $E_{\mathrm{Coulomb}}^{\mathrm{inter}}$ and $E_{\mathrm{hb}}^{\mathrm{inter}}$ yield only a small modification of the total intermolecular energy landscape dominated by $E_{\mathrm{vdW}}^{\mathrm{inter}}$. This implies that the role of hydrogen bonding may be less important than previously assumed. Hydrogen bonding has been considered as one of the main interactions for the cluster formation of *TrisAzo* and its variants [37,38].

Next, an energy decomposition is carried out based on the interactions between the different physical components of the two *TrisAzo* molecules. Due to their existence as separate chemical compounds and their distinct roles in the molecule, it is reasonable to distinguish three segments: the BTA core at the center (BTA), the three Azo groups attached to it (Azo), and the dimethylamino groups (DMA) at the periphery of the arms. Thus, $E_{\mathrm{tot}}^{\mathrm{inter}}$ of the dimer is decomposed into the following contributions:$E_{\mathrm{AA}}^{\mathrm{inter}}$ — Azo–Azo,$E_{\mathrm{BB}}^{\mathrm{inter}}$ — BTA–BTA,$E_{\mathrm{DD}}^{\mathrm{inter}}$ — DMA–DMA,$E_{\mathrm{AB}}^{\mathrm{inter}}$ — Azo–BTA,$E_{\mathrm{AD}}^{\mathrm{inter}}$ — Azo–DMA,$E_{\mathrm{BD}}^{\mathrm{inter}}$ — BTA–DMA.

These contributions again sum to the total intermolecular energy. The energy terms for analogous components of the molecules are symmetric, e.g., $E_{\mathrm{AA}}^{\mathrm{inter}}=E\left({\mathrm{Azo}}^{1},{\mathrm{Azo}}^{2}\right)$ for molecules 1 and 2. The terms involving different parts of *TrisAzo* are defined in a symmetrized manner as well. For instance, $E_{\mathrm{AB}}^{\mathrm{inter}}=E^{\mathrm{inter}}\left({\mathrm{Azo}}^{1},{\mathrm{BTA}}^{2}\right)+E^{\mathrm{inter}}\left({\mathrm{BTA}}^{1},{\mathrm{Azo}}^{2}\right)$.

The resulting energy landscapes are presented in Figure 4. The Azo–Azo interactions yield the strongest attractive contributions of all considered interactions. Pronounced negative energies down to $-23.4\;k\mathrm{cal}\;\cdot \;\mathrm{mol}{}^{-1}$ are found at $\Delta \varphi =0{}^{\circ }$, $120{}^{\circ }$, and $240{}^{\circ }$. At in-between rotation angles, the interactions are slightly repulsive but close to zero. Hence, $E_{\mathrm{AA}}^{\mathrm{inter}}$ is responsible for the $C_{3}$-symmetrical pattern of the potential wells in $E_{\mathrm{tot}}^{\mathrm{inter}}$. The dominance of $E_{\mathrm{AA}}^{\mathrm{inter}}$ underlines that dispersion interactions between stacked Azo groups (π–π interactions) have the largest impact on the intermolecular energy of the dimer. Likewise, the Azo–BTA interactions yield attractive contributions of a comparable magnitude. The values of $E_{\mathrm{AB}}^{\mathrm{inter}}$ are overall negative and on the order of $-10\;k\mathrm{cal}\;\cdot \;\mathrm{mol}{}^{-1}$ (with $\min E_{\mathrm{AB}}^{\mathrm{inter}}=-16.2\;k\mathrm{cal}\;\cdot \;\mathrm{mol}{}^{-1}$).

At Δr≳3.0 Å the BTA–BTA interactions are slightly attractive for all rotation angles. Thus, the interactions between the BTA linkers promote the binding of the dimers, but less strongly than the Azo–Azo and Azo–BTA interactions. Note that $E_{\mathrm{BB}}^{\mathrm{inter}}$ includes the explicit H-bonding contributions (Figure 3d) and dispersion interactions since the two phenyl rings can interact via π–π stacking. Figure 4b suggests that the steric repulsion stems only from the BTA–BTA interactions. This results from constraining the atom positions of the phenyl ring at the center of the BTAs. Atoms belonging to other parts of the molecules can avoid close distances during the energy minimization because they are not constrained, while the central carbon atoms of the BTAs are fixed (see Section 2.4 for details). Moreover, the obtained optimal stacking distances for the BTAs and the *TrisAzo* dimers are overall on the order of 3.7–4.0 Å. This is in good agreement with reported stacking distances in columnar supramolecular assemblies, particularly, in assemblies containing π–π-stacked and hydrogen-bonded BTAs [13,64].

Lastly, the peripheral dimethylamino groups contribute only marginally to $E_{\mathrm{tot}}^{\mathrm{inter}}$. Only the Azo–DMA interactions are slightly attractive. Meanwhile, $E_{\mathrm{DD}}^{\mathrm{inter}}$ and $E_{\mathrm{BD}}^{\mathrm{inter}}$ are minimally repulsive but overall close to zero. Especially for $E_{\mathrm{BD}}^{\mathrm{inter}}$, the distance between the two components is very large in the considered dimer geometries.

Using this decomposition by the molecule components, one may estimate the intermolecular energy landscape of a *TrisAzo-H* dimer, i.e., a dimer of the *TrisAzo* variant with R = H, as used in the experiments of Lee et al. [37]. The single hydrogen atoms, now replacing the dimethylamino groups, have a negligible effect on $E_{\mathrm{tot}}^{\mathrm{inter}}$. The total intermolecular energy of *TrisAzo-H* is then given by $E_{\mathrm{AA}}^{\mathrm{inter}}+E_{\mathrm{BB}}^{\mathrm{inter}}+E_{\mathrm{AB}}^{\mathrm{inter}}$. The resulting energy landscape is qualitatively almost identical to the one of *TrisAzo* but with energy minima that are shallower by about $-10\;k\mathrm{cal}\;\cdot \;\mathrm{mol}{}^{-1}$. Thus, the supramolecular aggregation of *TrisAzo* and *TrisAzo-H* should be similar, energetically speaking. Note, that the presence of the bulkier dimethylamino groups in *TrisAzo* leads to a larger excluded volume of the molecule.

### 3.2. Pre-Assembly of Columnar TrisAzo Clusters

Next, the structural properties of larger *TrisAzo* clusters solvated in water are examined under equilibrium conditions and compared with the findings regarding the binding in *TrisAzo* dimers. Initially, a set of pre-assembled columnar aggregates of varying sizes and structures is prepared. Every *TrisAzo* molecule is present as an all-*trans* isomer, since external light irradiation is not considered. The number of monomers in the stacks ranges from N=2 to 36. Based on previous self-assembly simulations [46] and the results for the *TrisAzo* dimer, the neighboring molecules are initially stacked at a COM distance of Δr=4 Å. The internal arrangements of the pre-assembled clusters are varied to examine which geometries are promoting the stability of the columnar shape. Three types of initial structures, *I*–*III* (Figure 5a) are prepared based on considerations about their intermolecular binding mechanisms:*I* Linear arrangement—Columns are built from centrally aligned monomers ($\Delta \varphi =0{}^{\circ }$). Such an arrangement corresponds to a concatenation of *TrisAzo* dimers at the minimum of their intermolecular energy (Figure 3a). This structure maximizes the alignment between the Azo groups but may be unfavorable for hydrogen bond formation, for which consecutive in-plane rotations by $60{}^{\circ }$ between the BTAs would be optimal [14,15,18].*II* $60{}^{\circ }$ twist—In this case, columnar aggregates are built from centrally aligned *TrisAzo* molecules with an in-plane rotation of $\Delta \varphi =60{}^{\circ }$ between neighboring pairs. This is expected to maximize the formation of hydrogen bonds and may possibly enable the strongest binding between the BTA cores, as found for columnar stacks of BTAs [14,15,18]. However, this rotation leads to a maximum displacement between the Azo arms of consecutive molecules. According to the dimer results (Section 3.1), this geometry is energetically less favorable than type *I*, but by considering it the impact of H-bonds on the binding can be examined.*III* Alternating (flipped) linear arrangement—Here, every second molecule in the column is flipped by $\Delta \alpha_{ij}=180{}^{\circ }$, i.e., rotated around an axis perpendicular to the stacking axis. After the flip, the rotation angle $\Delta \varphi_{ij}$ is slightly adjusted to maximize the alignment of the Azo arms. This is equivalent to rotating all three Azo arms of every second molecule by $180{}^{\circ }$ in cluster type *I*. Such arm rotations have only a small energy barrier [15,16], and are, thus, occurring frequently for free *TrisAzo* molecules. In this cluster arrangement, all three hydrogen bonds between adjacent molecules are expected to form. The amides in the neighboring BTA cores are arranged in such a manner, that a simple rotation of these groups results in their H-bonding. At the same time, the Azo groups are aligned as well, which is expected to yield a strong binding. The difference between this arrangement and type *I* can be seen most clearly from the positioning of the amide groups, e.g., the amide oxygens (red), in Figure 5a.

All aggregates are placed in rectangular simulation boxes with dimensions of at least $L_{x}\times L_{y}\times L_{z}=100\;Å\times 100\;Å\times 128\;Å$ ($L_{z}=200$ for N=36). The columnar axis of the stacks is oriented along the *z*-direction. Then, the aggregates are solvated in water and equilibrated for 10 ns. Thermal equilibrium is established when selected energetic (total energy, pair energy) and structural properties (next neighbor distances, columnar orientation parameter) are stable. Subsequently, the production runs follow for 25 ns.

### 3.3. Structure of the Clusters after Equilibration

Figure 5b depicts the largest simulated assemblies of each type (N=36) at the end of the production runs. None of the clusters disassemble into spatially separate parts, but clear differences in their shape are apparent from visual inspection. Whereas the clusters of type *III* preserve their nearly ideal columnar ordering, the other types are deviating from this by various degrees. The clusters of types *I* and *II* are compartmentalized into column-like sections, which are shifted or tilted with respect to each other. On top of that, these clusters appear curved and have lost their previously elongated and columnar shape.

Let us examine the inner structure of selected assemblies in detail. At first, only the largest stacks (N=36) are considered before taking into account different cluster sizes. The here computed observables are described in Appendix A. Note that monomers pairs range from i=1 to N−1 and monomers are indexed in a consecutive manner (*i*, i+1 instead of *i* and *j*). Figure 6 depicts the time-averaged local orientational order parameter ${\langle \Psi_{i}\rangle }_{t}$ and the time-averaged pairwise distances ${\langle \Delta r_{i}\rangle }_{t}$ of the stack as a function of the pair index *i*.

For the cluster of type *III*, both the orientational order and the pairwise distances are very stable at values close to $\Psi_{i}\approx 1.0$ and $\Delta r_{i}\approx 4\;Å$, respectively (Figure 6c,f). Hence, this assembly remains in almost perfect columnar alignment for nearly all neighboring monomer pairs. Only one of the two terminal *TrisAzo* pairs (i=35) has an increased COM distance and is slightly misaligned. This is because molecules at the end of the columnar stack bind only to one side, as opposed to the “inner” molecules, that bind to two direct neighbors.

Large irregularities occur in the clusters of types *I* and *II* (Figure 6a,b,d,e). In both cases, only small sections of two to four consecutive pairs remain in a column-like arrangement. These sections are interrupted by pairs with significant spatial and orientational deviations, especially for type *II*. For example, six pairs in the type-*II* cluster have values ${\langle \Psi_{i}\rangle }_{t}<0.5$ (inclination angles $\Delta \alpha_{i}>60{}^{\circ }$). By contrast, for the linearly arranged clusters of type *I* only two such pairs exist. In some instances, also a large spread of the data is observed. This indicates the formation or shifting of defects in these positions.

The in-plane rotation angles between the stacked molecules paint a similar picture as the structural parameters discussed above, see Appendix A. In clusters of type *III*, these angles deviate only to a small extent from the initial geometries. By contrast, clusters of type *I* occasionally show large in-plane rotations inside the stack but with Δφ being below $90{}^{\circ }$. In type-*II* clusters, the original helical twist of $\Delta \varphi =60{}^{\circ }$ is nearly never maintained. Instead, angles in the whole available range are found, but with a majority being close to $0{}^{\circ }$.

Next, we examine the structural properties of equilibrated clusters with different sizes. Figure 7a,b depict the average pairwise distances, ${\langle \Delta r\rangle }_{t}$, and the global columnar orientation order parameter, ${\langle \Psi \rangle }_{t}$, respectively, for all considered cluster types and sizes.

Small clusters with N≤6 have similar values of ${\langle \Delta r\rangle }_{t}$ independent of the cluster type. The pairwise distances show a slight increase from about 3.7 Å for N=2 to slightly above 4 Å for N=6. The standard deviation of Δr is much larger at N=6 than at N=2 and 3. In addition, the order of the columnar alignment is very stable for all clusters of sizes N≤3 with ${\langle \Psi \left(t\right)\rangle }_{t}≳0.95$. The smallest clusters (dimers, trimers) have, therefore, almost identical structural properties, independently of their initial arrangement, and have adopted a common, stable geometry.

In larger clusters (N>6), structural differences appear depending on the cluster type. Within clusters of types *I* and *II*, the average next-neighbor distances increase with *N*. The values of ${\langle \Delta r\rangle }_{t}$ reach the order of 5 Å and show large standard deviations. The alignment between the monomer pairs also decreases, with ${\langle \Psi \left(t\right)\rangle }_{t}$ dropping below 0.9 or sometimes 0.8 and a large spread in the data. Meanwhile, the number of defects in these clusters (Figure 7c) strongly increases with the size, roughly following a linear trend. Accordingly, the spatial correlations between the orientation unit vectors ${\vec{u}}_{i}$ and ${\vec{u}}_{j}$ decrease rapidly as the distance |j−i| of the monomer pairs increases (Figure 7d). The decay is slightly stronger for type *II*, than for type *I*. These results confirm a coexistence of some tightly stacked and some rather dislocated monomer pairs within these clusters (Figure 6b,c).

By contrast, as the columns of type *III* increase in their length, the pairwise distances decrease to around 3.7–4.0 Å with only small standard deviations. The orientational alignment is also nearly perfect with ${\langle \Psi \left(t\right)\rangle }_{t}≳0.95$ and the orientational correlations persist over long distances |j−i| (Figure 7d). In fact, these correlations decay only as |j−i|→N−1, i.e., between the pairs near the ends of the column. For type-*III* clusters, no defects are detected (Figure 7c). The deviations at the ends of the columns are below the applied thresholds as well. Thus, clusters of type *III* demonstrate a tight homogeneous stacking and remain in near perfect columnar order. Long columnar stacks of this type appear to be even more structurally stable than short ones. Note that in large clusters, the influence of the terminal *TrisAzo* on the average values is reduced.

### 3.4. Intermolecular Energy of the Clusters after Equilibration

#### 3.4.1. Total Intermolecular Energy of the Clusters

To develop an understanding of the structural properties, the intermolecular (or binding) energies between the molecules are considered. The total intermolecular energy of a cluster, $E_{\mathrm{tot}}^{\mathrm{inter}}$, is computed as the sum of the intermolecular energy contributions between all possible pairs of molecules in the assembly, i.e., not only between directly neighboring pairs. Figure 8a depicts $\langle E_{\mathrm{tot}}^{\mathrm{inter}}\rangle$ for the various cluster sizes and types. The intermolecular energies are normalized by the number of directly neighboring pairs in the cluster, N−1. This removes the expected increase of the binding strength as a function of the cluster size. Thus, changes in the binding energy per neighboring pair upon increasing the cluster size may be revealed. Even in this normalized representation, the intermolecular energies decrease nearly monotonically with increasing *N* for all cluster types. Hence, all considered cluster types are characterized by cooperative binding. For very large cluster sizes, the energy curves appear to approach plateaus. In the case of the strongly ordered type-*III* clusters, such a saturation sets in very early, i.e., already for intermediate cluster sizes (N=6). The binding energies of the other cluster types would reach their supposed limiting values only beyond the considered values of *N*.

For short columns (N<6), the binding energies after equilibration are very similar for all three cluster types. This correlates with the very similar structural properties for these small assemblies (Figure 7). For larger clusters (N≥6), the total intermolecular energies of the different cluster types are ordered III>II>I, when comparing them at constant values of *N*. Interestingly, clusters of type *III*, which are characterized by high columnar order, show binding energies of smaller magnitude than the disordered assemblies of type *I* and *II*. Their very regular shape also implies a smaller entropy. Note, however, that the columnar cluster structure of type *III* is stabilized by stronger anisotropic forces arising from hydrogen bonding (see further below).

#### 3.4.2. Decomposition of the Intermolecular Energy of the Clusters

More detailed information about the binding in the various cluster types can be obtained by again considering the separate intermolecular contributions. To this end, the intermolecular energies are decomposed based on

the contributions of each monomer;the types of non-covalent interactions; andthe interactions between different parts of the molecule.

The decomposition by contributions of individual monomers or monomer pairs is carried out as follows. The intermolecular energy $E_{i,j}^{\mathrm{inter}}$ between a pair of *TrisAzo* molecules *i* and *j* is obtained by only considering interactions of atoms that belong to these two molecules while excluding intramolecular interactions. Due to symmetry, it holds that $E_{i,j}^{\mathrm{inter}}=E_{j,i}^{\mathrm{inter}}$. The contributions can therefore be arranged in a symmetric matrix:(3)$$E_{i,j}^{\mathrm{inter}}=\left(\begin{array}{cccc} E_{1,1}^{\mathrm{inter}} & E_{1,2}^{\mathrm{inter}} & \cdots & E_{1,j}^{\mathrm{inter}}\\ E_{2,1}^{\mathrm{inter}} & E_{2,2}^{\mathrm{inter}} & \cdots & E_{2,j}^{\mathrm{inter}}\\ \vdots & \vdots & \ddots & \vdots \\ E_{i,1}^{\mathrm{inter}} & E_{i,2}^{\mathrm{inter}} & \cdots & E_{i,j}^{\mathrm{inter}} \end{array}\right).$$

Note that the sum over all unique pairs (i≠j) equals the total intermolecular energy: $\sum_{i\neq j}E_{i,j}^{\mathrm{inter}}=E_{\mathrm{tot}}^{\mathrm{inter}}$.

Using the matrix representation, the non-covalent interactions between the monomers can be analyzed in various ways. Let us consider how to obtain the average intermolecular energies of monomers with separations |j−i| along the stack (Figure 8b). For any value |j−i|, the values $\langle E_{i,i+|j-i|}^{\mathrm{inter}}\rangle$ are extracted from the matrix $E_{i,j}^{\mathrm{inter}}$ by averaging the entries of the |j−i|th off-diagonal above the main diagonal. As expected, the interactions between directly neighboring monomers are the strongest. Molecules that are separated by more than |j−i|=1 along the column are contributing attractively as well, i.e., they further stabilize the cluster. The strength of the intermolecular energies decays steeply upon increasing their separation |j−i|. In cluster types *I* and *II* also fairly separated pairs interact. This is mainly a consequence of the structural irregularities observed in these cluster types, that bring some molecules closer together than in a perfectly ordered columnar stack. The strongest attractive contributions for |j−i|>1 stem from dispersion interactions (inset of Figure 8b) because the intermolecular Coulomb energies are generally of a lesser magnitude.

Next, $E_{\mathrm{tot}}^{\mathrm{inter}}$ is decomposed into the different non-covalent contributions (Figure 9). Overall, the results are very similar to the energy decomposition of the *TrisAzo* dimers. For all cluster types and sizes, attractive van der Waals (dispersion) contributions dominate the total intermolecular energy. For the all considered clusters, the intermolecular Coulomb interactions and hydrogen bonds promote the binding as well, but by a much smaller amount. Among the two, $E_{\mathrm{hb}}^{\mathrm{inter}}$ is generally more pronounced than $E_{\mathrm{Coulomb}}^{\mathrm{inter}}$. Together, $E_{\mathrm{Coulomb}}^{\mathrm{inter}}$ and $E_{\mathrm{hb}}^{\mathrm{inter}}$ amount to only about 10–20% of $E_{\mathrm{vdW}}^{\mathrm{inter}}$ in these systems. All three cluster types have a very similar intermolecular energy composition for N=2 because of their very similar equilibrium structures (Figure 7). For larger clusters, the less-ordered cluster types *I* and *II* have nearly identical energy contributions as well. By contrast, in long type-*III* clusters, the van der Waals contribution is noticeably weaker and the H-bonding energy is about doubled, as compared with types *I* and *II*. Hydrogen bonding, hence, plays a stronger role in these well-ordered columns (type *III*) but is still weaker than the dispersion interactions.

Lastly, $E_{\mathrm{tot}}^{\mathrm{inter}}$ is decomposed into the interactions between the different components of the individual *TrisAzo* molecules, see Figure 10. Note that the energy decomposition is carried out in a generalized manner, as compared with the energy decomposition for the *TrisAzo* dimers (Section 3.1). For instance, the Azo–BTA contribution $E_{\mathrm{AB}}^{\mathrm{inter}}$, here, contains all intermolecular interactions of the Azo groups of a certain molecule *i* with the BTA cores of all other molecules *j* in the cluster, summed over all *i*.

For N=2 (Figure 10a), the results for all three considered cluster types are consistent with the intermolecular energies of *TrisAzo* dimers (Section 3.1). Most interactions are attractive and among them, the Azo–Azo contributions are the most pronounced, followed by BTA–BTA and Azo–BTA. The Azo–DMA interactions are attractive as well but with an intermediate strength. By contrast, the DMA–DMA and BTA–DMA interactions are negligible due to the large distances between these parts, their weaker interactions, and the smaller number of atoms involved.

In larger clusters, here N=18 (Figure 10b), the intermolecular energies are slightly different. The strongest and weakest interactions remain the same but for types *I* and *II*, the Azo–BTA, and Azo–DMA contributions become much more attractive. This is a result of their irregular arrangement, leading to increased contacts between these “mixed” parts of neighboring monomers. In the very regular stacks of type *III*, the Azo–BTA and Azo–DMA remain similar as in the dimers, whereas the BTA–BTA interactions are becoming more attractive. This is consistent with the stronger H-bonding interactions and the stable central alignment of the monomers in these stacks.

### 3.5. The Role of Hydrogen Bonding

Besides the energetic contributions of the hydrogen bonds, their number and pattern are important to consider. H-bond formation strongly depends on meeting specific geometrical criteria between the opposing amide groups, i.e., short distances and favorable inclination angles. Therefore, the number and pattern of the hydrogen bonds contain information about the alignment between the monomers and, in particular, their central parts. The H-bonding in the clusters is determined by analyzing the MD trajectories via a customized version of the Hydrogen Bond Analysis module of the Python library MDAnalysis [66,67]. The algorithm detects a hydrogen bond if the donor (nitrogen), hydrogen and acceptor (oxygen) atoms form an angle larger than $120{}^{\circ }$ and the distance between the hydrogen and the acceptor atom is below 3 Å. Figure 11a shows the average number of H-bonds, $\langle n_{\mathrm{hb}}\rangle$, for the three considered cluster types. The values of $\langle n_{\mathrm{hb}}\rangle$ are normalized by the number of adjacent pairs in the column, N−1, to obtain an average measure of H-bonding between the directly neighboring BTA groups.

For N≤6, the three cluster types behave very similarly with about 2 to 2.5 hydrogen bonds per pair. These values are slightly below the three possible H-bonds between each BTA pair, indicating a moderately large number of hydrogen bonds. This result is in line with the similar structural properties of short stacks, independent of their initial arrangement. For clusters with N>6, the three cluster types strongly differ in their H-bond count. The very regular clusters of type *III* have almost the ideal number of three H-bonds per stacked BTA pair. This implies that on average all three amide groups in each stacked *TrisAzo* molecule are involved in hydrogen bonding. Appendix A demonstrates the very regular succession of H-bonds in these stacks and Figure 11b visualizes the helical, triple H-bonding pattern. The orientation of each amide group has not been analyzed for the full trajectories. However, one can see that the cluster section in Figure 11b is characterized by an asymmetric 2:1 pattern for each *TrisAzo* pair. The presence of a triple-hydrogen-bonding pattern is in agreement with the known H-bonding between BTA dimers and has been found in other studies on BTA stacks [15,18]. Note that each amide group of an “inner” *TrisAzo* molecule in such stacks is H-bonded to an amide group in the molecules above and below. Long clusters of types *I* and *II*, which have been shown to be of a much less regular structure, maintain about two hydrogen bonds per pair. Hence, they do not form or keep up the ideal H-bonding pattern. Appendix A underline that the H-bonding pattern in these clusters is much more irregular. In particular, some well-aligned *TrisAzo* pairs in these clusters form three H-bonds, whereas fewer or no hydrogen bonds are found between other pairs. Due to the rotation of the amide groups upon hydrogen bonding, BTA stacks have been found develop large macrodipole moments, which can impact their intercolumnar interactions [13,15,16,64,68]. The formation of macrodipoles in the here-simulated *TrisAzo* stacks is dicussed in Appendix A.

What do these results imply about the role of hydrogen bonding in the *TrisAzo* stacks? Recall Figure 9, which demonstrates that the energetic contributions of H-bonds are much weaker than the van der Waals interactions in all considered stacks. Nevertheless, the maintenance of a very regular H-bonding pattern between the BTAs is strongly correlated with the stability of the columnar cluster shape. It is plausible that the regularity of both characteristics—the hydrogen bonding and the stack arrangement—is mutually dependent, rather than being directly related via cause and effect. Thus, H-bonds have a stabilizing role despite their comparatively minor energetic contributions.

Let us discuss the simulation results in relation to the experimental observations of Lee et al. [37,38]. The cited studies infer that hydrogen bonding in the observed aggregates occurs in two possible ways. First, H-bonds are formed between the amide groups of two adjacent *TrisAzo-H* molecules, as described above. Second, there may exist an additional solvent-mediated hydrogen bonding as prompted by DFT results [37]. The latter has not been observed in our simulations because the utilized model does not explicitly involve hydrogen bond formation for water molecules. Lee et al. conjecture that the typical triple hydrogen-bonding pattern between BTAs is unlikely to persist in the Azo star clusters due to the bulkiness of the azobenzene arms [37]. Our MD simulations demonstrate that, in fact, this triple-H-bonding pattern can be maintained but only in the alternating stack geometry (type *III*). Lastly, the experiments reported in Refs. [37,38] demonstrate that hydrogen bonding is essential for the stability of the supramolecular assemblies, in agreement with our results. In particular, it was reported that the deliberate destruction of H-bonds in the sample by adding fluoride anions leads to a collapse of the needle-shaped *TrisAzo* structures.

### 3.6. Rationalizing the Structural Differences of the Considered Cluster Types

After analyzing the structural properties and energy compositions of the different cluster types in equilibrium, the question remains why the different initial cluster arrangements lead to such pronounced structural differences after equilibration. To this end, we study the intermolecular interactions within the initial cluster structures. We aim to find out, whether the expectations for the intermolecular interactions in the initial cluster structures are valid (compare Section 3.2) and what this implies about the relevant binding mechanisms in the stacks. The intermolecular energies and hydrogen bonds of the clusters are extracted for the first 0.1 ns of the equilibration run. During this time, the clusters are virtually still present in their pre-arranged state. The results of this analysis are shown in Figure 12a–e.

An important consideration in selecting the pre-assembled structures was the assumption that strongly aligned Azo groups between neighboring molecules enhance the binding strength via pronounced dispersion interactions. This effect has been confirmed for all clusters in equilibrium and the same is expected for the initial assemblies of types *I* and *III*. For type-*II* clusters, the consecutive $60{}^{\circ }$ in-plane rotation angles should significantly weaken the dispersion interactions between neighboring monomers. Figure 12c,d confirm that the van der Waals energies of type-*II* clusters are strongly reduced as compared with the equilibrated clusters (Figure 9) or the other two initial arrangements. The reduction in $E_{\mathrm{VdW}}^{\mathrm{inter}}$ is directly associated with weaker Azo–Azo interactions in these clusters (Figure 12e), which confirms the hypothesized relation. In addition, also the Azo–DMA interactions are less attractive due to the non-aligned arms of neighboring monomers. As a consequence of these changes, the total intermolecular energies of type *II* are almost halved in comparison with types *I* and *III* at the beginning (Figure 12a). Due to the substantially weaker binding, the molecules in type-*II* clusters can rearrange themselves and diverge from their pre-assembled state. These findings illustrate the importance of the Azo stacking for the maintenance of very regular *TrisAzo* columns. By contrast, the initial energies of type-*I* and type-*III* clusters are already very similar to the equilibrium results (Figure 9 and Figure 10). Their final states, therefore, must differ for other reasons as detailed below.

Next, we consider the hydrogen bond formation in the three initial cluster arrangements. The initial cluster arrangements of types *II* and *III* are expected to be beneficial for the formation of H-bonds. In the type-*I* cluster geometry, the opposing amide groups are not as favorably positioned for hydrogen bonding, and thus, fewer H-bonds are expected. This conjecture is confirmed as well, see Figure 12b. Clusters of type *I* contain only about 1 to 1.5 hydrogen bonds per neighboring pair, depending on the cluster size. This is in strong contrast to the other two cluster types, which always form more than two H-bonds per neighboring pair. For N>2, the value of $\langle n_{\mathrm{hb}}\rangle$ is between 2 and 2.5 for type *II* and near 3 for type *III*. For N=2 the results are reversed. In the initial state, the intermolecular H-bonding energies of type *II* are much stronger than in the other cluster types (Figure 12c,d). For cluster types *I* and *III*, $E_{\mathrm{hb}}^{\mathrm{inter}}$ is about equal, at roughly half of the values of the type-*II* clusters. Accordingly, also the BTA–BTA energies of the latter clusters are more attractive (Figure 12e). These results reveal two things: First, H-bonding is not sufficient to maintain regular columnar stacks if the dispersion interactions between the Azo groups are effectively removed. For this reason, type-*II* clusters do not remain column-like, as discussed above. Second, the main difference in the binding of type-*I* and type-*III* clusters is the higher number and more regular pattern of H-bonds in the latter. Hence, together with strong dispersion interactions (π–π stacking) of the Azo arms and BTA cores, clusters of type *III* can stay in a well-ordered columnar shape. They also increase their H-bonding energies over time (Figure 9). Interestingly, the H-bonding energies of cluster types *I* and *III* are similar in the beginning. Hence, type-*III* clusters initially have more, but weaker H-bonds. This finding highlights that the regularity of the H-bonding pattern and the associated proximity and alignment of the BTA groups is effective for maintaining well-ordered stacks. By contrast, the presence of fewer but stronger hydrogen bonds is not sufficient.

Finally, it should be noted that in a real sample of *TrisAzo* molecules, the formation of aggregates most probably does not occur in either of the idealized pre-assembled ways considered in this investigation. During the self-assembly of the *TrisAzo* molecules into stacks, more degrees of freedom play a role than those that have been considered here for pre-arranged clusters. The simulation results mainly highlight the importance of specific interactions for the intracolumnar binding. In particular, the dominant contribution of the Azo groups and the stabilizing influence of H-bonds are standing out. It can be expected that these binding mechanisms will play a central role in real systems, as well.

## 4. Conclusions

This computational study analyzes the structural properties and the binding mechanism of supramolecular stacks formed by *TrisAzo* molecules. The stacks are investigated in fully atomistic MD simulations, since this method enables a detailed examination of the intermolecular interactions beyond experimentally achievable results.

We found that the stability of the cluster shape, including the tendency towards defects and other structural deviations, strongly depends on the initial arrangement of the monomers. The most structurally robust of the considered cluster arrangements (type *III*) is characterized by a strong alignment between the Azo groups of neighboring monomers, as well as nearly optimal hydrogen bond formation between the opposing BTA cores. Pre-stacked cluster arrangements that maximize only the Azo-group alignment (type *I*) or the H-bonding between the BTAs (type *II*) degrade over time and lose their columnar arrangement. While the study focuses on rather idealized stacking scenarios, our results indicate that the experimentally observed aggregates of Azo stars are expected to have highly aligned Azo groups [37,38]. Note, that despite the strongly fluctuating cluster structures, there has been no spontaneous breaking or detachment of cluster parts within the simulation time.

Decomposing the intermolecular energies of the clusters reveals that supramolecular aggregates of *TrisAzo* molecules are held together by specific intermolecular interactions. The strongest contributions to the binding of the stacks stem from intermolecular interactions between the opposing Azo groups, between the Azo groups and BTA cores, and between the stacked BTA cores (in this order by magnitude). The peripheral dimethylamino groups have only a minor influence on the binding. Thus, the results of this study may be applied to stacks of *TrisAzo-H* molecules. Decomposing the intermolecular energies with respect to the non-covalent interactions reveals that the binding is dominated by van der Waals interactions (dispersion). This can be attributed to the π–π stacking between the conjugated parts of the molecules—particularly, between the Azo groups and the central BTA cores. Additional attractive interactions stem from hydrogen bonds between the amides in the BTA cores but are far weaker. Nonetheless, the presence of a very regular triple H-bonding pattern is highly correlated with a stable columnar shape of the cluster. The comparison between the initial and final clusters underlines the roles of the Azo groups and the hydrogen-bonded BTAs in these systems. The dominance of the Azo–Azo interactions also suppresses the helical arrangement of the molecular building blocks. The average in-plane rotations angles between consecutive monomers are found to be below the value of $60{}^{\circ }$, which would be expected for BTA stacks [14,15,18].

The presented investigation provides a quantitative examination of the different intermolecular interactions in the columnar assemblies, which has not been addressed in our previous works on this topic [46,47]. In particular, we quantify how the different non-covalent interactions and the functional groups of *TrisAzo* contribute to the binding energy of the stacks. We demonstrate how the interplay of these different energetic contributions influences the equilibrium structure of the stacks. Thus, we provide a deeper comprehension of the structure formation in this system.

Since the study focuses on the properties of the aggregates in equilibrium, it serves as a characterization of the system in its reference state, i.e., before the application of light. Another work using the same force field representation follows this publication [69]. There, we examine how *TrisAzo* stacks respond to light that triggers the photoisomerization of azobenzene. In total, the two works deepen the understanding of supramolecular assemblies, incorporating photoswitchable groups. Thereby, we may aid the future design of light-responsive materials based on supramolecular structures, such as gels and supramolecular polymers.

## Figures and Tables

**Figure 1 molecules-26-07598-f001:**
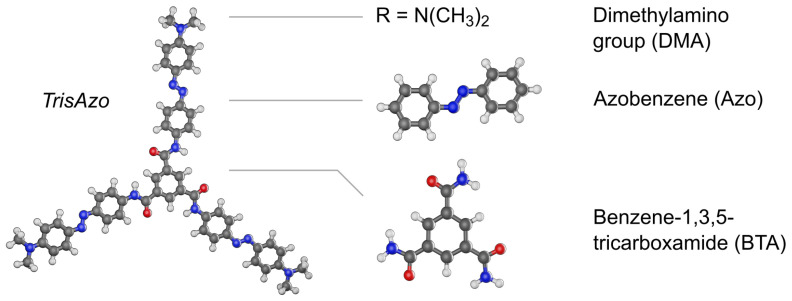
The star-shaped multiphotochromic molecule *TrisAzo* and its constituents. The compound consists of three azobenzene (Azo) groups, each para-substituted with a dimethylamino (DMA) group, and linked at the center by BTA.

**Figure 2 molecules-26-07598-f002:**
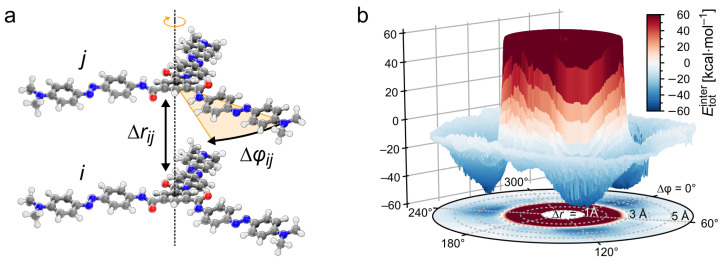
(**a**) Geometrical variables considered for the analysis of the intermolecular energy: pairwise distance $\Delta r_{ij}$ and rotation angle $\Delta \varphi_{ij}$ between a pair *i* and *j* of *TrisAzo*. (**b**) Three-dimensional energy landscape (total intermolecular energy) of a *TrisAzo* dimer. The energies are depicted in a range of colors: blue—negative, grey—zero, red—positive. Positive energy values beyond $60\;k\mathrm{cal}\;\cdot \;\mathrm{mol}{}^{-1}$ are not shown. The absence of data for Δr<1 Å is indicated by an empty (white) area.

**Figure 3 molecules-26-07598-f003:**
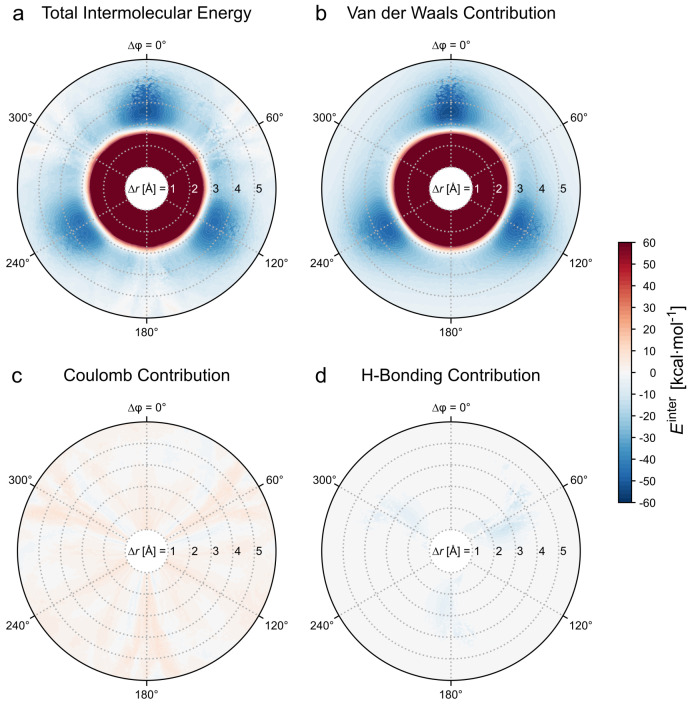
(**a**) Total intermolecular energy of a *TrisAzo* dimer and (**b**–**d**) its decomposition into contributions due to non-covalent interactions: (**b**) van der Waals interactions, (**c**) Coulomb interactions, and (**d**) hydrogen bonding. All plots employ the same energy scale. The color scheme corresponds to the one in Figure 2b. The empty (white) area at Δr<1 Å indicates the absence of data in this distance range.

**Figure 4 molecules-26-07598-f004:**
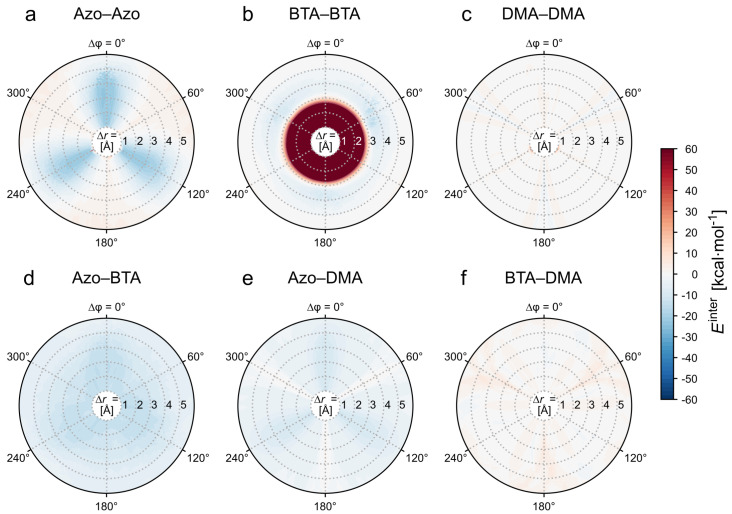
Decomposition of the intermolecular energy of a *TrisAzo* dimer by the interactions arising between different parts of the molecules. Upper row: Interactions between corresponding components, (**a**) Azo–Azo, (**b**) BTA–BTA, and (**c**) DMA–DMA. Lower row: Interactions between mixed components, (**d**) Azo–BTA, (**e**) Azo–DMA, and (**f**) BTA–DMA. All plots use the same energy scale. The color scheme corresponds to the one in Figure 2b. The absence of data for Δr<1 Å is indicated by an empty (white) area.

**Figure 5 molecules-26-07598-f005:**
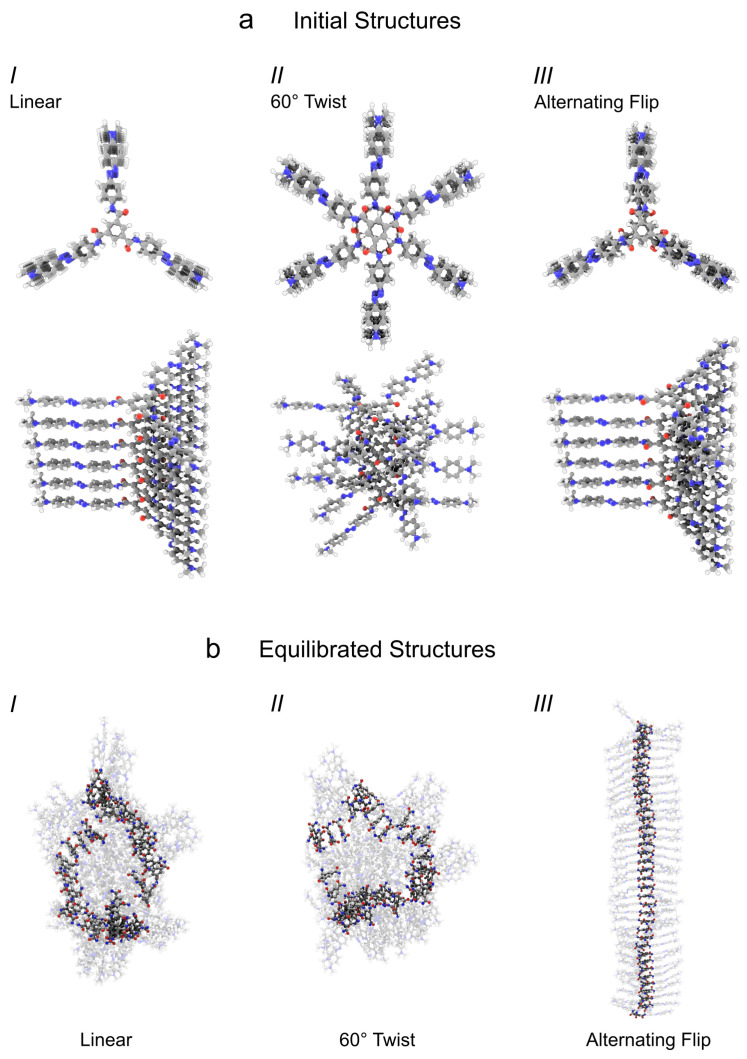
(**a**) The three considered initial structures of the pre-assembled columnar *TrisAzo* clusters, here with N=6 (top and angled view). From left to right: *I*—linear arrangement, *II*—$60{}^{\circ }$ twist (or in-plane rotation) between neighboring molecules, and *III*—linear arrangement with alternating molecules flipped by $180{}^{\circ }$. (**b**) Simulation snapshots of the resulting equilibrium geometries, taken at the end of the production runs. The largest considered clusters are shown (N=36). The BTA linkers are highlighted for better visibility of aligned sections and defects.

**Figure 6 molecules-26-07598-f006:**
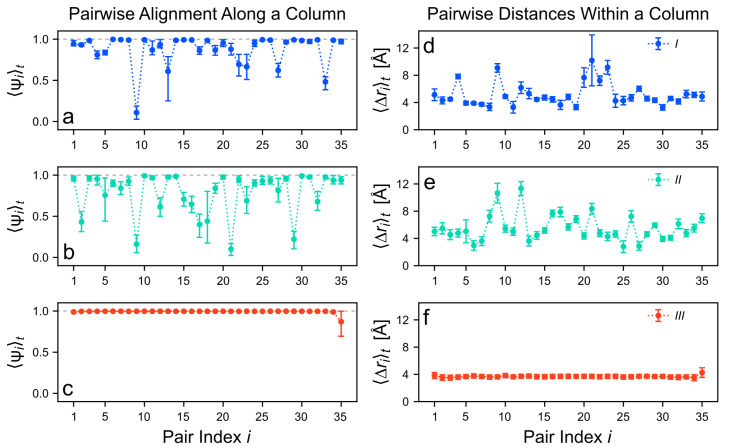
(**a**–**c**) Time averages of the local columnar orientation order parameter, ${\langle \Psi_{i}\rangle }_{t}$. (**d**–**f**) Time averages of the pairwise distances, ${\langle \Delta r_{i}\rangle }_{t}$. Both observables are plotted as a function of the pair index *i* along the stack. The largest simulated clusters (N=36) are shown for each cluster type. The error bars in (**a**–**c**) delimit 68% confidence intervals. In (**d**–**f**) standard deviation error bars are used. Dashed gray lines in (**a**–**c**) indicate the largest possible value of ${\langle \Psi_{i}\rangle }_{t}$. Dotted lines between the symbols serve as a guide to the eye.

**Figure 7 molecules-26-07598-f007:**
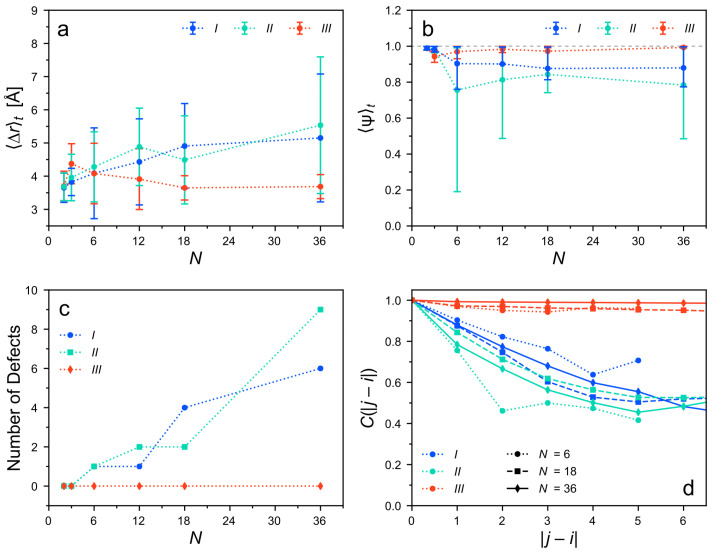
Structural properties of the different cluster types as a function of the cluster size. (**a**) Average pairwise distance and (**b**) average columnar orientation order parameter. The error bars in (**a**) denote the standard deviation and in (**b**) delimit 68% confidence intervals. (**c**) Number of defects in the clusters. (**d**) Correlation function of the orientation unit vectors ${\vec{u}}_{i}$ and ${\vec{u}}_{j}$ as a function of the monomer separation |j−i| along the stack [65]. The different cluster types are distinguished by colors. Dotted lines serve as a guide to the eye.

**Figure 8 molecules-26-07598-f008:**
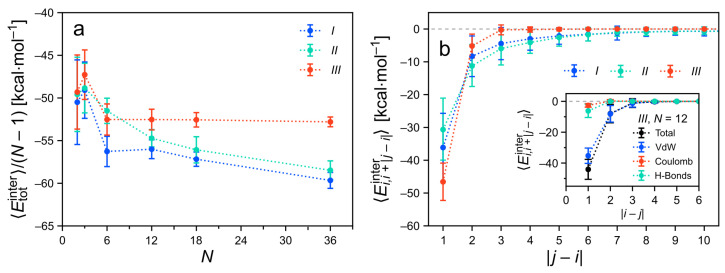
(**a**) Intermolecular energies $E_{\mathrm{tot}}^{\mathrm{inter}}$ per neighboring pair of monomers as a function of the cluster size *N*. (**b**) Average intermolecular energy between monomer pairs at separations |j−i| along the stack. Here, only clusters with N>10 are considered. The inset shows the contributions of the different non-covalent interactions to $E_{i,i+|j-i|}^{\mathrm{inter}}$ for a specific cluster (type *III*, N=12).

**Figure 9 molecules-26-07598-f009:**
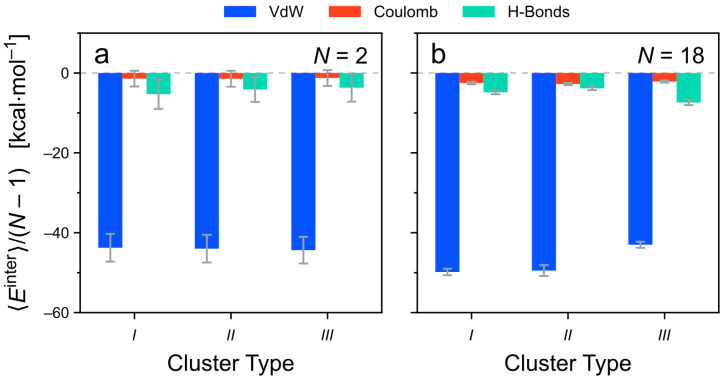
Contributions of the different non-covalent interactions to the intermolecular energy of the *TrisAzo* stacks. The energies are divided by the number of neighboring monomer pairs (N−1) for better comparability. Results for (**a**) N=2 and (**b**) N=18 are shown for each cluster type.

**Figure 10 molecules-26-07598-f010:**
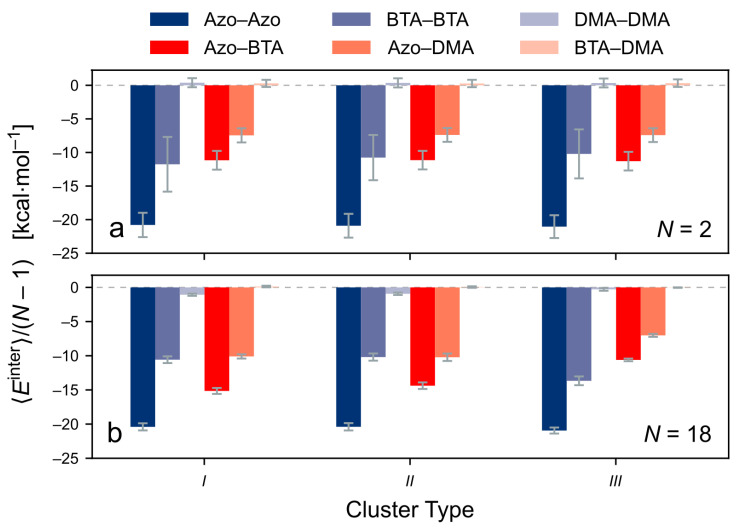
Decomposition of the intermolecular energy of the clusters into contributions due to interactions between the various physical parts of the *TrisAzo* molecules. The energies are divided by the number of neighboring monomer pairs (N−1) for better comparability. Results for (**a**) N=2 and (**b**) N=18 are shown for each cluster type.

**Figure 11 molecules-26-07598-f011:**
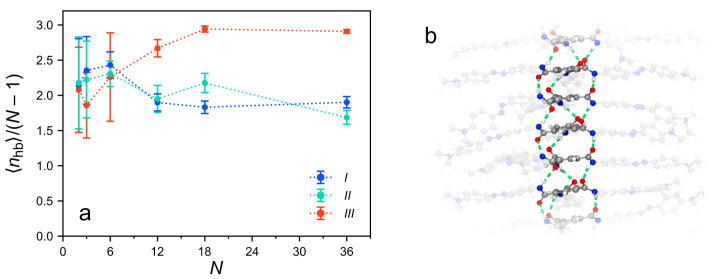
(**a**) Average number of hydrogen bonds per neighboring pair as a function of the cluster size *N*. Different cluster types are distinguished by color. Dotted lines serve as a guide to the eye. (**b**) Detail of an MD snapshot showing a *TrisAzo* stack (cluster type *III*, N=18) with indicated hydrogen bonds (dashed green lines). The BTA centers are highlighted to improve visibility. The full snapshot is shown in Appendix A.

**Figure 12 molecules-26-07598-f012:**
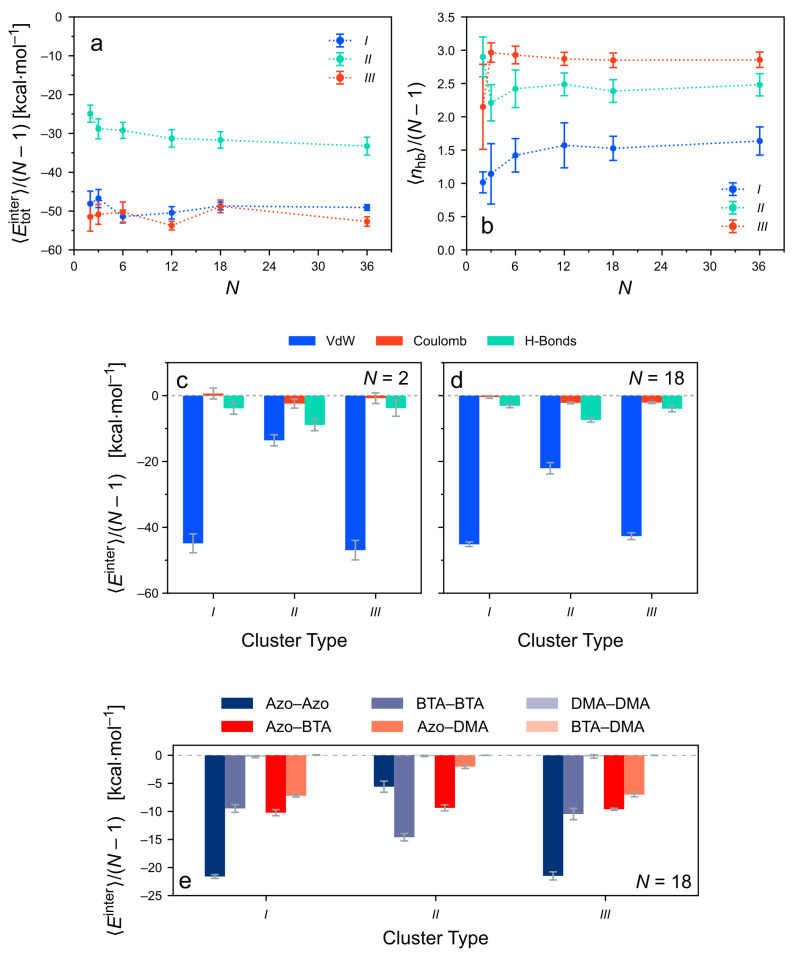
Intermolecular energies and hydrogen bonding in the *TrisAzo* stacks in their initial state (first 0.1 ns of the equilibration run). (**a**) Total intermolecular energies, (**b**) average number of hydrogen bonds per neighboring pair, (**c**,**d**) decomposition of the intermolecular energy by types of non-covalent interactions, and (**e**) decomposition of the intermolecular energy by interactions between different molecule parts. Dashed and dotted lines serve as a guide to the eye.

## Data Availability

The data presented in this study are available on request from the corresponding authors.

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
