# Peer review of "Columnar Aggregates of Azobenzene Stars: Exploring Intermolecular Interactions, Structure, and Stability in Atomistic Simulations"

_molecules, 2021, doi:10.3390/molecules26247598_

Round 1

Reviewer 1 Report

In this manuscript, computational study is devoted to structural properties and the binding mechanism of supramolecular stacks formed by TrisAzo molecules. I recommend for the publication of the manuscript after the following concerns and suggestions are considered.

  1. In section 2.3, the authors stated that “the intermolecular energies of numerous 140 dimer configurations are calculated, each providing a data point to the energy landscape.” However, the details of the calculation are still absent. I don’t know how to compute the total interaction energies.
  2. The decomposition of total interaction energies leads to three terms, vdW, Coulomb and hb. The definitions of these contributions should be provided.

Reviewer 2 Report

The submitted manuscript named [Columnar Aggregates of Azobenzene Stars: Exploring Intermolecular Interactions, Structure, and Stability in Atomistic Simulations] is a computational study of self-assembled structure, which is formed by specific three-arm azobenzene star molecules (named TrisAzo) in water. Evaluation of this paper was not easy:

  • Positive features: The authors performed professional, detailed and sophisticated computational study. The manuscript is written in good and understandable language.
  • Negative features: The manuscript contains a lot of calculation results, but no experimental data. The previous papers of the group around the main author published two quite similar studies in the past (references [46, 47] in the submitted manuscript). The main difference from between this manuscript and references [46, 47] seems to be different computation method (as the authors admit in the 7th paragraph of Introduction).
  • Conclusion: From formal point of view, the manuscript seems to be correct and all I can do is to recommend some [minor revisions]. From the point of view of originality and novelty, it is a up to editor’s decision if he/she decides to accept a manuscript to a journal with IF = 4.4 if the results are similar (although not identical to) the previous studies of the authors. Maybe it would help if the authors could briefly explain, which are the new aspects of their study (see also my specific comments below).

Specific comments:

(1) Abstract: (a) The authors talk about experimental results, but the paper is a pure theoretical/computational study, which is confusing. The authors should say something like “PREVIOUS experimental results showed…”. (b) In line 11, the authors say: We FIND that binding energies of the stacks are dominated by π-π interactions, but in fact they just CONFIRMED this fact, which had already been reported in their previous studies. Moreover, the fact that planar aromatic molecules are stabilized mostly by π-π interactions is quite expectable. I suggest that the authors somehow modify the text so that this was clearer.

(2) Introduction, 2nd paragraph: Potential and real (i.e. realized) applications of supramolecular assemblies are mixed. Please, separate them clearly into two groups: potential applications and real applications.

(3) The manuscript is rather long. I suggest that the authors focused on key results and moved as much text as reasonable to Appendixes and Supplementary materials.

(4) Results and discussion: I suggest that the authors inserted a brief subsection BRIEFLY listing and explaining the new findings with respect to their previous studies (namely references [46,47]).
